# Towards Automation of Knowledge Graph Generation

Rishi Midha[1][0000−0002−2494−6314] and Enayat Rajabi[1][0000−0002−9557−0043]

Shannon School of Business, Cape Breton University, Sydney NS, Canada
{cbu19ffj,enayat_rajabi}@cbu.ca

**Abstract.** As the incredible promise behind knowledge graphs becomes evident through commercial success, it is clear why they were touted as the wave of the future. The primary goal of this study is to investigate the challenges we faced during the automatic creation process of a domain-independent knowledge graph for open statistical data. Through the creation of said graph, this study identifies and enlists the pitfalls that impede the realization of automatic construction of knowledge graphs.

**Keywords:** Knowledge graph · Semantic web · Linked open data.

## 1 Introduction

The adoption of Knowledge Graph by Google in 2012 to amplify the search engine's capabilities led to mainstream appeal and brought attention to the field of semantic web development. An interesting area of application is open statistical data elicited from sources including but not limited to government portals and public repositories. However, almost a decade later, due to sporadic research in the field, its potential still remains untapped. The majority of open statistical data are published by governments to be used and commercialized by a wide range of users. With the integration of open statistical data from multiple sources, linking them to each other and to the external ontologies and datasets, and creating semantic rules a knowledge graph can be constructed. This knowledge graph can enable the performance of advanced data analytics and facilitate answering more sophisticated queries.

In this paper, we investigate the automatic construction of a knowledge graph for open statistical data. Section 2 discusses knowledge graph construction techniques and the related studies in this domain. Section 3 describes the method we followed to construct a knowledge graph for open statistical datasets. Discussion and results of this investigation are provided in Section 4. The final section concludes this research by stating the findings.

## 2 Related Works

Different studies have developed various ontologies and knowledge graphs to achieve knowledge management and knowledge sharing systems [3], [1]. They

have built large-scale, high-quality, general-purpose knowledge bases, such as YAGO, DBpedia, Wikidata, CN-DBpedia. [1] analyzed the construction approaches of the knowledge graph in spacecraft launch in terms of knowledge source, modeling, extraction, fusion, inference, and storage. A knowledge graph is constructed based on ternary representation form of "entity-relationship-entity" or "entity-attribute-value". The knowledge graph nodes usually represent real-world entities and the edges show semantic relationships between these entities. A knowledge graph can be constructed in a) top-down approach where the entities are added to the knowledge-base based on a predefined ontology, or b) bottom-up approaches where knowledge instances are extracted from knowledge base systems and then, the top-level ontologies are built based on the knowledge instances to create the whole knowledge graph [3]. [2] proposed automatic approach for constructing a knowledge graph in healthcare, where by combining manual labeling and automatic relationship extraction model. In this study, we followed the top-down approach to construct a knowledge graph on open statistical data, where we leverage the advantage of semantic web technologies to build a knowledge-based system.

## 3   Methods

A knowledge graph construction process can be performed based on the following steps: a)Knowledge acquisition: Collecting semi-structured data from an API, b) Knowledge extraction: Extracting entities and their relationships, c) Knowledge fusion: Constructing an ontology, assigning entities and relationships to the knowledge graph based on the defined ontology, and interlinking entities to external ontologies and datasets, and d) Knowledge storage: Storing the created knowledge graph in a triple store

The study involves generating a knowledge graph for open statistical data and sources this data primarily from open statistical data portals. As Figure1 illustrates, the workflow commences with the ingestion of datasets and structure them in the form of entities, relationships, and semantic rules using a custom ontology. The main component of this process is the knowledge graph constructor, where it uses the designed ontology, an interlinking component to link data to external ontologies and datasets, and ingested datasets to generate the final knowledge graph.

The following subsections describe different components of this process in details:

### 3.1   Data collection

The majority of open data portals expose their datasets via Scorata API [1]. For this study, we retrieved datasets from open Nova Scotia[2] data portal using Python[3]. It is here that the data is fitted onto the custom ontology created

---

[1] https://dev.socrata.com/

[2] https://data.novascotia.ca

[3] https://www.python.org

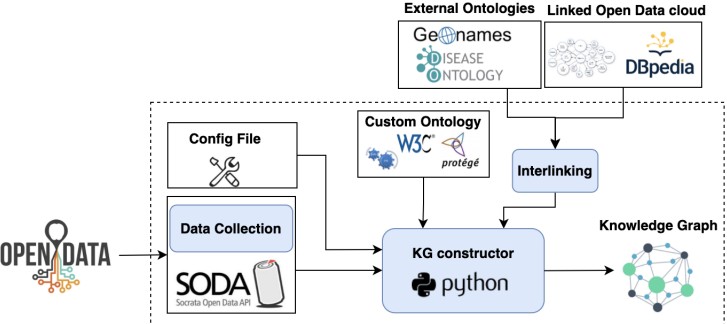

**Fig. 1.** Knowledge graph construction process for open statistical data

using Protege tool[4]. The knowledge graph is then created using a customised code which is compatible with domain-independent open statistical data. The data is enriched with extensive linking and the final graph is available for users to perform advanced data analytics.

### 3.2 Ontology

The ontology for open statistical datasets was developed based on the W3C standard and the best practice vocabularies such as RDFS, SDMX, dcterms, VoID, RDF cube, and SKOS. The external ontologies such as DOID and GeoNames have been used as well to enrich the knowledge graph with domain knowledge. The datasets were coded as entities with distinct data structure definitions, slices and observations. The data structure definitions contain information about the dimensions and measures that apply to all observations or records of the datasets. The custom features to model the web portal such as category and theme are coded using DCAT vocabulary to further promote gold-standard of linked open data.

### 3.3 Knowledge Graph Construction

The knowledge graph constructor is created using python libraries RDFLib, OWLReady2, and Graph. Using the custom ontology developed, the datasets are modelled into triples and parsed onto the graphs. Given the lack of structural metadata about the dimensions and measures of the datasets, we developed a configuration setting to specify the dimensions, measures and thier links them to industry standards. This allows semi-automatic updating of the graph with input data and makes the datasets semantically and dimensionally connected to each other, to the external ontologies and to the linked open data cloud. The semantic web rules language is also translated into the constructor component to enable semantic reasoning over the knowledge graph. Finally the datasets

---

[4] https://protege.stanford.edu

are added onto the graph as observations and it is ensured that they conform to prescribed metadata, structure and semantic web protocols. The knowledge graph itself is checked against well-received works of the field in terms of concept, schema, entity instances and relations.

## 4   Discussion

The study aims to present an end-to-end solution to transform domain-independent open statistical data into knowledge graph. Due to certain limitations which have been identified below, there is hindrance in the attainment of complete automation.

When transforming datasets into triples using W3C recommendations, an integral part is defining the data structure definition which contains details about dimensions and measures thus defining the structure of the dataset and its observations. Lack of descriptive structural metadata that enlists the dimensions, measures, and attributes of each dataset explicitly is the biggest hurdle towards achieving complete automation. Alternatively, the lack of a vocabulary that supports properties that convey this information is another issue which prevents us from addressing it in a standardized manner. A workaround presented in this paper is the use of a configuration file that contains mapping details for dimensions and measures which shows promise for said datasets but is insufficient in regards to scalability.

Another problem is the lack of consistency across datasets in terms of structure and nomenclature. The structural consistency refers to strict rules defining the attributes in terms of atomicity and datatypes. For instance, it is best practice to express fields such as address in a partitioned fashion comprising of various elements such as post code, street address to simplify information retrieval. The nomenclature for certain common dimensions and measures is not regularised which further causes challenges in mapping.

Finally, the lack of specialized tools dedicated to creation and management of knowledge graphs and other semantic web concepts proves to be inefficient and detrimental for research. As an example, the design of custom ontology and interlinking are completed on Protege, but this tool fails to be efficient for data ingestion at scale via APIs. A pioneering next step in this field would be development of an end-to-end solutions that combine best practices and leverages industry standards to overhaul this process of knowledge graph creation. With the added user-friendliness and easy access as well as efficiency, this will help churn out more semantic web resources and enrich the entire eco-system as whole through collaboration and integration.

## 5   Conclusion

In this paper, we outlined the steps we followed and the challenges towards automating the creation of knowledge graph for open statistical data.

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
