# OpenReview forum: "Towards Automation of Knowledge Graph Generation"
_eswc-conferences.org/ESWC/2021/Conference/Poster_and_Demo_Track — Submitted to ESWC2021 P&D_

### Official Review · AnonReviewer2 · 2021-04-08
**Interesting topic but the paper scientific contribution is not clear**

**Rating:** 5
**Confidence:** 5

**Review:**

The paper describes a methodology for the automatical creation of knowledge graphs from open statistical data.
The topic is interesting and challenging. However, the authors' details are poor and there are not references to other works that can explain some steps. The description is vague. From the paper, I understand that the author's only relevant contribution is the usage of a configuration file that provides mapping information for dimensions and measures.

There is not evaluation/case study that proves the effectiveness of the proposed approach. There is only a section about current limits.

In other words, it is not clear the focus/contribution of the paper.

Minor issues:
- Please, check carefully the paper for mistakes
- pag.2 "As Figure1" -> "As Figure 1"

**Anonymity:**

Yes, I would like my review to remain anonymous.

---

### Official Review · ~Yoan_Chabot1 · 2021-04-11
**More focus should be given to the specificities of the data considered**

**Rating:** 2
**Confidence:** 5

**Review:**

This paper presents a method for automatically creating a knowledge graph from statistical data.
The problem (although interesting and highly strategic) and the proposed approach are presented in a manner that is not specific enough to allow readers to appreciate the real contributions of the work.
The authors should put much more emphasis on the specificities of their work through different actions proposed below.

First of all, the keywords used at the beginning of the paper are too generic for a conference like ESWC.
"Knowledge Graph", "Semantic Web" and "Linked Open Data" are keywords that provide too little information.
I suggest that the authors rework these keywords and the abstract to allow readers to understand early on what the paper is about and its contributions.

Then, the introduction fails to introduce the problem and the inherent challenges.
The growing adoption of knowledge graphs and research into automatic KG construction methods is a very interesting topic but should not constitute the whole introduction of the paper.
The authors need to present the challenges inherent in the statistical data which seem to be one of the particularities of this research work.
I suggest reworking the introduction by presenting this specific problem, the related challenges and the contributions of the work to address them.
Furthermore, the authors should better motivate the interest of constructing a knowledge graph from statistical data.
Possible applications such as "advanced data analytics" and "facilitate answering more sophisticated queries" should be specified.

The next section introduces existing work on the automatic construction of knowledge graphs
The paper would benefit if the section was more oriented towards statistical data and its particularities.
Are there already efforts to build KG on statistical data? How does the proposed work compare with what already exists? These are all questions to which the authors should provide answers.
Are there specific modelling issues when considering statistical data? If so, which ontologies can meet this need, even partially?

The third section presents the process of converting the datasets into a knowledge graph.
The proposed method is a very classical method of knowledge graph construction.
The authors do not mention the specificities of statistical data and it is difficult for the reader to perceive the nature of the data and the interest of its transformation into a knowledge graph.

An ontology has been developed to structure the knowledge graph. The tools and ontologies used as references are presented in detail but no word is given on the dimensions represented in the ontology.
I invite the authors to rework this section by presenting the important dimensions of the statistical data considered which should be as many concepts and relations in the ontology (a figure presenting the ontology could help the readers to visualise these semantic dimensions)

The extraction of knowledge from the datasets to feed the knowledge graph is then described in a sentence that is not very informative and does not allow to understand how the process is actually conducted:
"Using the custom ontology developed, the datasets are modelled into triples and parsed onto the graphs".
More details on the techniques used should be provided in order to show how the approach handles the specificities of the statistical data.
For this, the authors need to explain in a more precise paragraph how the "KG constructor" module works.
Beyond the libraries, what are the techniques used? (OTTR rules? Ad-hoc mappings?)

Regarding the form, a better balance should be found between the "Discussion" and "Conclusion" sections.
The current conclusion does not add anything to the paper.

The paper should be consistent in spacing:
* On the API and Python footnotes on page 2
* In the lists on page 2, "a)Knowledge" VS "b) Knowledge
* Space missing: "As Figure1" on page 2

Typos: "thier" (page 3)

**Anonymity:**

No, I would like my review to be deanonymized.

---

### Official Review · AnonReviewer3 · 2021-04-12
**It is difficult to verify the main outcomes from the text of the paper**

**Rating:** 3
**Confidence:** 4

**Review:**

The paper describes an approach to automate a KG construction. The authors leverage statistical data from open data source(s) (https://data.novascotia.ca/ is mentioned in the paper) and generate an RDF cube dataset. In addition, authors interlink the data to LOD sources such as DBpedia and Geonames.

The **main goal** of the paper as stated by the authors is "identify and enlist the **pitfalls** that impede the realization of automatic construction of knowledge graphs". Unfortunately, the paper does not describe the details of the process beyond Figure 1 and some additional remarks in text -- for example, usage of specific APIs, RDF vocabularies and Python frameworks. Therefore, it is difficult to verify the claims of the authors.

Among those **pitfalls** authors mention:

1. ````"*Lack of descriptive structural metadata that enlists the dimensions, measures, and attributes of each dataset*". In order to verify this it would be useful to know further details of the data itself and the parameters of the ontology. The authors mention that they use RDF cube and SDMX, but it remains unclear why concepts offered by SDMX vocabulary (for example, here: https://raw.githubusercontent.com/UKGovLD/publishing-statistical-data/master/specs/src/main/vocab/sdmx-metadata.ttl) is not enough for their purposes. Moreover, the authors mention a lack of vocabularies, however, there is not discussions which vocabularies were considered and why those do not fit the requirements.
2. "*lack of consistency across datasets in terms of structure and nomenclature*". This problem seems a known barrier for industry applications. It is not clear which approaches authors have considered to overcome this difficulty. For example, did you try such tools as Silk (http://silkframework.org/)? Or tools from this work: Nentwig, Markus, et al. "A survey of current link discovery frameworks." Semantic Web 8.3 (2017): 419-436.
3.  "*the lack of specialized tools dedicated to creation and management of knowledge graphs*". With respect to this point authors mention Protege that is an ontology modelling tool. It appears to me that ETL tools could be more suitable for this task, for example, UnifiedViews (https://github.com/UnifiedViews), LinkedWidgets (https://publik.tuwien.ac.at/files/PubDat_235029.pdf), Linked Pipes (https://etl.linkedpipes.com/).

A more detailed description of the data processing and, therefore, more details about the faced challenges would also help to make the main contributions more targeted, and therefore, more fruitful / original.

In Section 3.3 **Knowledge Graph Construction** authors code their own solution in Python. If I understand correctly, the data is first parsed into Python base structure (string, integers, etc.) and then a graph is constructed from this data. It would be useful to consider alternative approaches such as using (R2)RML to restructure the data or even OBDA to allow ontology based querying of the data.

The **related work** section lists 3 industry specific implementations KG construction / curation systems. I would suggest to extend the overview considering at least the following papers:

- The authors are concerned with semantic intergration of data, including discovery and alignment of duplicates. Diego Collarana, Mikhail Galkin, Ignacio Traverso-Ribón, Maria-Esther Vidal, Christoph Lange, and Sören Auer. 2017. MINTE: semantically integrating RDF graphs.
- This approach uses OBDA techniques to query virtual RDF graphs. De Giacomo G., Lembo D., Lenzerini M., Poggi A., Rosati R. (2018) Using Ontologies for Semantic Data Integration.
- In this paper authors also consider data integration into a RDF data cube dataset. K. Koupidis, C. Bratsas, S. Karampatakis, A. Martzopoulou and I. Antoniou, "Fiscal Knowledge discovery in Municipalities of Athens and Thessaloniki via Linked Open Data,
- The authors consider a scalable approach for data integration Mami M.N., Grangel-González I., Graux D., Elezi E., Lösch F. (2020) Semantic Data Integration for the SMT Manufacturing Process Using SANSA Stack. In: Harth A. et al. (eds) The Semantic Web: ESWC 2020 Satellite Events. ESWC 2020.


Additional remarks:

- Is ontology publicly available?
- It is better to not start a sentence with a reference (see Related Work).
- Minor typos in the text.
- The references are formatted incorrectly. Please, include a full list of authors for each paper.

**Anonymity:**

Yes, I would like my review to remain anonymous.

---

### Decision · Program_Chairs · 2021-04-19

Reject